# Defective memory engram reactivation underlies impaired fear memory recall in Fragile X syndrome

Jie Li, Rena Y Jiang, Kristin L Arendt, Yu-Tien Hsu, Sophia R Zhai, Lu Chen*

Department of Neurosurgery, Department of Psychiatry and Behavioral Sciences, Stanford University School of Medicine, Stanford, United States

**Abstract** Fragile X syndrome (FXS) is an X chromosome-linked disease associated with severe intellectual disabilities. Previous studies using the *Fmr1* knockout (KO) mouse, an FXS mouse model, have attributed behavioral deficits to synaptic dysfunctions. However, how functional deficits at neural network level lead to abnormal behavioral learning remains unexplored. Here, we show that the efficacy of hippocampal engram reactivation is reduced in *Fmr1* KO mice performing contextual fear memory recall. Experiencing an enriched environment (EE) prior to learning improved the engram reactivation efficacy and rescued memory recall in the *Fmr1* KO mice. In addition, chemogenetically inhibiting EE-engaged neurons in CA1 reverses the rescue effect of EE on memory recall. Thus, our results suggest that inappropriate engram reactivation underlies cognitive deficits in FXS, and enriched environment may rescue cognitive deficits by improving network activation accuracy.

## Introduction

Fragile X syndrome (FXS) is a neurodevelopmental disorder associated with intellectual disability and a leading genetic cause of syndromic autism spectrum disorders (ASDs) (*Santoro et al., 2012*). FXS is caused by loss-of-function mutations in the fragile X mental retardation 1 (*FMR1*) gene, which encodes the fragile X mental retardation protein (FMRP) (*Verkerk et al., 1991*). The most common cause of FXS is the expansion of CGG trinucleotide repeats within the 5′ untranslated region of the *FMR1* gene, which triggers hypermethylation of the CpG island in the promoter region and subsequent silencing of the *FMR1* gene, resulting in the loss of FMRP expression (*Fu et al., 1991*). Other loss-of-function mutations in the *FMR1* coding region have also been identified in rare cases of FXS (*Suhl and Warren, 2015*). As an RNA-binding protein, FMRP interacts with and regulates the translation of a large number of mRNAs encoding proteins involved in regulation of neuronal morphology and synaptic function (*Ascano et al., 2012*). Thus, in the absence of FMRP, dysregulated protein synthesis affects multiple neuronal pathways, generating behavioral phenotypes displayed by FXS patients, such as repetitive behavior and impaired cognitive functions and social interactions (*Bagni et al., 2012*).

A FXS mouse model with constitutive deletion of the *Fmr1* gene (*Fmr1* KO) has been used extensively to study neural mechanisms underlying cognitive impairments of FXS patients (*Bakker et al., 1994*). Although results and conclusions from some of the behavioral tests conducted in these animals remain controversial (*Kazdoba et al., 2014*), the *Fmr1* KO mouse nonetheless recapitulates most of the behavioral characteristics displayed by FXS patients, including deficits in learning and memory (*Krueger et al., 2011*; *Santos et al., 2014*). Synaptic plasticity changes are often found to underly learning deficits in animal models of neurological diseases. Indeed, *Fmr1* KO mice exhibit various synaptic dysfunctions associated with FMRP deletion (*Contractor et al., 2015*; *Pfeiffer and Huber, 2009*), including excessive group 1 mGluR-dependent long-term depression (mGluR-LTD)

*For correspondence:
luchen1@stanford.edu

(*Hou et al., 2006*; *Huber et al., 2002*; *Koekkoek et al., 2005*), abnormal long-term potentiation (LTP) (*Lauterborn et al., 2007*; *Martin et al., 2016*; *Paradee et al., 1999*; *Tian et al., 2017*), and impaired homeostatic synaptic plasticity (*Sarti et al., 2013*; *Soden and Chen, 2010*; *Zhong et al., 2018*). Some of these findings, such as impaired homeostatic synaptic plasticity, has been validated in human neurons differentiated from FXS patient iPS cells (*Zhang et al., 2018*), indicating that they are translationally relevant. These results provide a strong correlation between altered synaptic function and memory deficits in FXS and predict that network activation patterns relevant to memory formation may be affected by synaptic dysfunction, thus contributing to learning deficits exhibited in behaving animals. In this study, we test this hypothesis by directly examining neural ensemble activation during learning and its reactivation during memory recall, and how accuracy of memory engram activation correlates with behavioral performance in normal and FXS mice.

## Results

### FMRP expression in hippocampal CA1 is required for contextual memory formation

Intellectual impairment is one of the behavioral features observed in FXS patients. Cognitive deficit has been extensively validated in *Fmr1* KO mice. In our hands, deficits in hippocampus-dependent learning in the *Fmr1* KO mice is evident in classical contextual fear conditioning (*Figure 1A and B*, *Figure 1—figure supplement 1A*) and passive avoidance tests (*Figure 1C and D*). We adapted an intensive training paradigm for the fear conditioning previously used in activity-dependent labeling studies (*Liu et al., 2012*) (four tone-foot shock pairs per session, three sessions spaced by 2 hr). In both behavioral tasks, *Fmr1* KO mice exhibited normal learning, indicated by their normal performance during and immediately after training (*Figure 1A and C*, *Figure 1—figure supplement 1A*). By contrast, their memory recall 1–3 days after training was significantly impaired in both tasks (*Figure 1B and D*). Hot plate test revealed no deficits in pain perception in the *Fmr1* KO mice (*Figure 1—figure supplement 1B*), suggesting that impaired memory is not likely due to compromised pain perception. Additionally, *Fmr1* KO mice performed similarly to their WT littermates in Y maze, open field test, elevated plus maze and novel object recognition test (*Figure 1—figure supplement 1C–F*), indicating that their working memory, locomotion, non-social anxiety levels and hippocampus-independent recognition memory are all normal.

Next, we probed specific contribution of FMRP expression in the hippocampal CA1 region to overall performance in hippocampus-dependent learning tasks. CA1-specific FMRP deletion was achieved with stereotaxic injection of Cre recombinase-expressing adeno-associated viruses (AAV-Syn-Cre-GFP) into the CA1 region of *Fmr1* conditional knockout (cKO) mice hippocampus 2–3 weeks before fear conditioning (*Figure 1E*, *Figure 1—figure supplement 1G*; *Zhong et al., 2018*). AAVs expressing an inactive truncated form of Cre (mCre) were used as a control. The CA1-*Fmr1* KO mice showed normal performance during and after training of conventional fear conditioning paradigm (one session of three tone-foot shock pairs) (*Figure 1F*, *Figure 1—figure supplement 1H*), but exhibited contextual memory deficits tested one day after training (*Figure 1G*). No deficit on pain perception, working memory, locomotion, anxiety, and object recognition memory were observed (*Figure 1—figure supplement 1I–L*). Taken together, these results indicate that normal expression of FMRP in hippocampal CA1 is required for contextual memory formation.

### Engram reactivation is impaired in *Fmr1* KO mice

The selective impairment of freezing during memory recall but not immediately after training in *Fmr1* KO mice prompted us to explore whether reactivation of memory-encoding engram cell ensembles during memory recall is altered. We utilized an activity-dependent tagging virus AAV-*Fos*-ER$^{T2}$-Cre-ER$^{T2}$-PEST (AAV-*Fos*-TRAP, a viral version of 'targeted recombination in active populations' [TRAP]) (*Guenthner et al., 2013*; *Ye et al., 2016*), which drives Cre expression under the control of the *Fos* promoter and in the presence of tamoxifen (*Figure 2—figure supplement 1A*). Injection of the AAV-*Fos*-TRAP in the tdTomato reporter mice Ai9 enables labeling of populations of neurons activated by a particular experience within a defined time window (*Figure 2—figure supplement 1B*). This genetic labeling method is highly specific and sensitive. Mice received vehicle injection showed minimal labeling, suggesting minimal expression leakage (*Figure 2—figure*

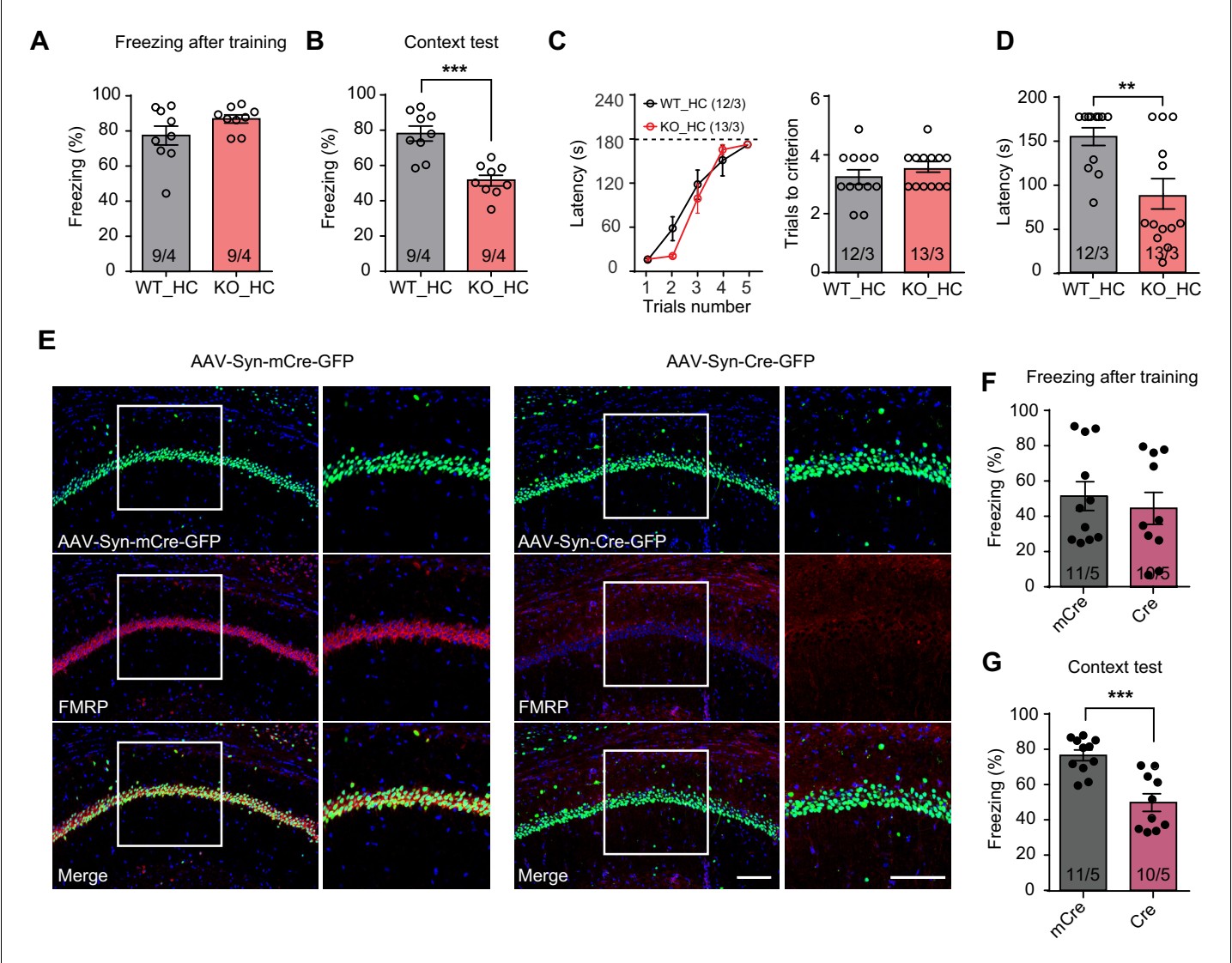

**Figure 1.** Impaired hippocampus-dependent contextual fear memory in constitutive *Fmr1* KO and CA1-*Fmr1* KO mice. (A and B) Contextual fear memory test in constitutive *Fmr1* KO mice. Freezing levels were measured immediately after fear conditioning training (A) and again in training context three days after (B) (***p<0.001, two-tailed unpaired *t* test). (C–D) Passive avoidance test in constitutive *Fmr1* KO mice. (C) Learning curves (left) and trials number to criterion (right) were measured during training, and contextual fear memory in passive avoidance was measured 1 day later (D) (**p<0.01, two-tailed Mann-Whitney *U* test). (E) Representative images of hippocampal CA1 regions from *Fmr1* conditional KO mice with bilateral injections of AAVs expressing Cre-GFP or mCre-GFP (green). FMRP expression was evaluated with immunostaining (red). Scale bars: 100 μm. (F and G) Contextual fear memory test in CA1-*Fmr1* KO mice. Freezing levels were measured immediately after fear conditioning training (F) and again in training context 1 day after (G) (***p<0.001, two-tailed unpaired *t* test). n/N, number of mice/number of independent litters. All graphs represent mean ± SEM. The online version of this article includes the following figure supplement(s) for figure 1:

**Figure supplement 1.** Behavioral characterization of constitutive *Fmr1* KO mice and CA1-*Fmr1* KO mice reared in home cage.

*supplement 1C and D*). Moreover, a 4 hr experience of an enriched environment (EE) (with a single dose of 4-Hydroxytamoxifen (4-OHT) at the beginning of EE) significantly increased tdTomato-labeled neurons in CA1 compared to control mice that were maintained in the home cage with basal neural activity (*Figure 2—figure supplement 1C and D*), indicating successful capturing of activated neuronal population by our method. Additional analysis indicates that majority of AAV-*Fos*-TRAP labeled neurons within the CA1 pyramidal layer are excitatory neurons as greater than 94% of them are positive for CaMKII staining (*Figure 2—figure supplement 1E and F*).

Having validated the labeling method, we next sought to examine engram reactivation in hippo-campal CA1 during fear memory recall. Learning-activated neurons were labeled with 4-OHT injection. Since the effective time window of 4-OHT is around 6 hr after injection (*Guenthner et al., 2013*), and the intensive training protocol spans more than 6 hr, we performed two 4-OHT injections at the beginning of the first and third training sessions to ensure sufficient labeling (*Figure 2A*). Contextual fear memory was tested 3 days after training to allow sufficient time for tdTomato expression. Hippocampal tissues were collected 1 hr after memory recall and processed for FOS immunohistochemistry staining to label recall-activated neurons (*Figure 2B*). Both fear conditioning training and memory recall activated comparable numbers of neurons in CA1 in both WT and *Fmr1* KO mice, indicated by similar percentage of tdTomato-expressing (training-activated) and FOS-positive (memory recall-activated) neurons in these mice (*Figure 2C and D*). Additionally, compared to tdTomato-negative neurons, a significantly higher portion of the tdTomato-positive neurons expressed FOS in both WT and KO mice, suggesting that learning-labeled memory engrams were

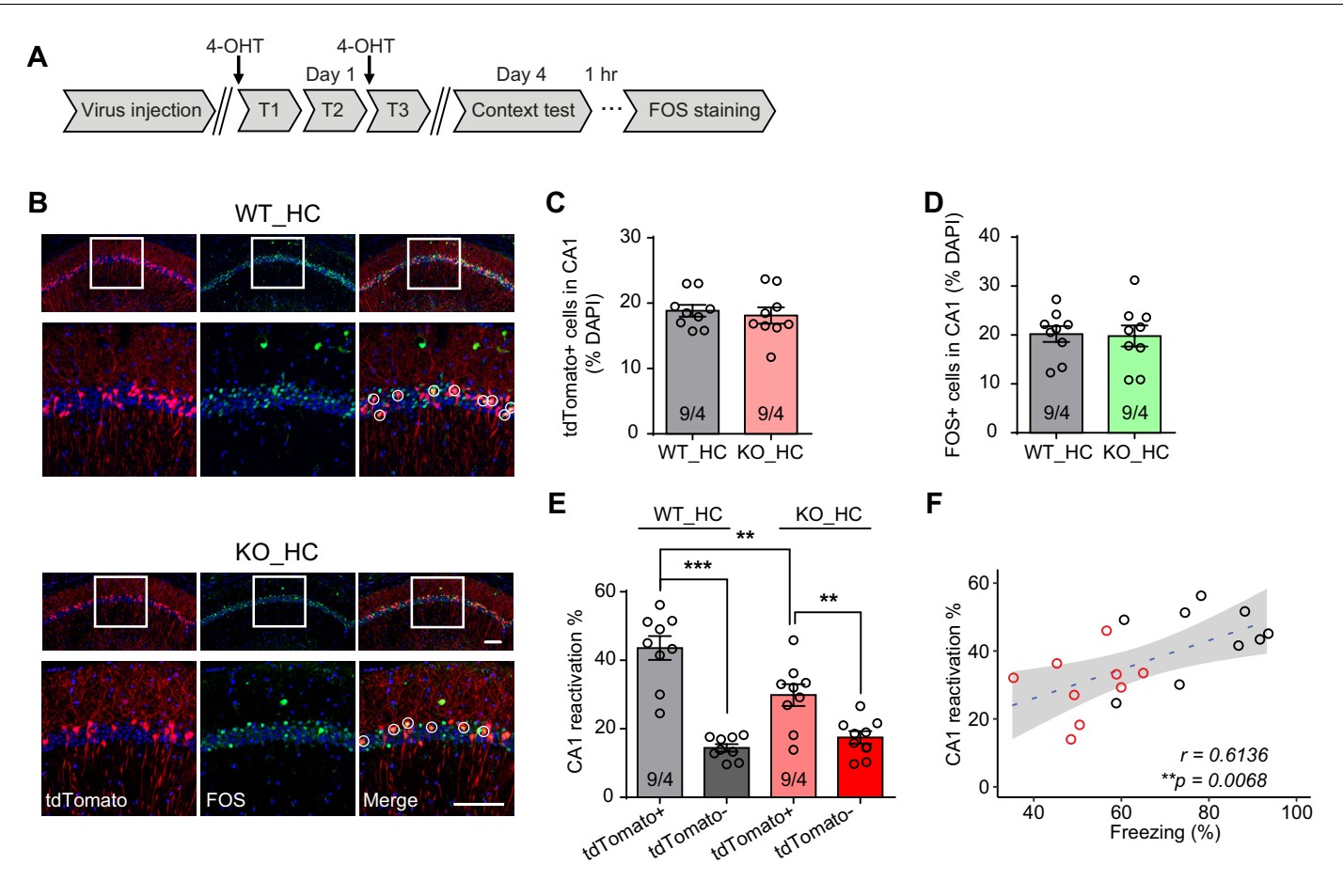

**Figure 2.** Reduced engram reactivation efficacy is correlated with impaired contextual fear memory in *Fmr1* KO mice. (A) Experimental protocol for activity-dependent genetic labeling of neural ensembles. Intensive fear conditioning training was used in constitutive *Fmr1* KO mice to facilitate labeling. (B) Representative images showing memory-encoding neural ensembles (engram cells) labeled with tdTomato (red) and memory recall-activated neurons labeled with FOS immunostaining (green). The circles in zoomed in images highlight reactivated neurons (yellow). Scale bar: 100 μm. (C) Quantification of percentage of neurons activated during learning. (D) Quantification of percentage of neurons activated during memory recall. (E) Quantification of engram reactivation efficacy in CA1 as percent FOS-positive neurons in tdTomato-positive and tdTomato-negative populations [one-way ANOVA with Tukey's multiple comparison test: $F$ (3, 32)=26.57, p<0.0001, ***p<0.001; **p<0.01]. (F) Positive correlation between neural ensemble reactivation efficacy and behavioral perform during context memory test (**p<0.01, Pearson correlation coefficient). n/N, number of mice/number of independent litters. All graphs represent mean ± SEM.

The online version of this article includes the following figure supplement(s) for figure 2:

**Figure supplement 1.** Validation of activity-dependent genetic labeling method and characterization of *Fos*-TRAP-labeled neurons.

preferentially reactivated during memory recall, consistent with their predicted roles in memory encoding (*Figure 2E*). However, compared to WT, this preferential reactivation is significantly reduced in *Fmr1* KO mice (*Figure 2E*), suggesting that reactivation efficacy is decreased in the *Fmr1* KO mice. Further, the Pearson correlation analysis of individual animal's freezing levels during recall and their CA1 engram reactivation efficacy showed a significant correlation between the two measurements (*Figure 2F*), indicating that CA1 engram reactivation efficacy is positively correlated with animal's performance during context memory test.

## Experiencing an enriched environment prior to learning rescues fear memory formation in *Fmr1* KO mice

Enriched environment (EE) experience, which provides rich sensory, motor, and social stimulation to mice, has been shown to improve cognitive function and ameliorate some symptoms of neurodevelopmental disorders (*Nithianantharajah and Hannan, 2006*). In FXS mice, EE has been shown to correct dendritic morphological abnormalities and rescue behavioral phenotypes of hyperactivity, social deficits and performances in some cognitive tasks (*Oddi et al., 2015*; *Restivo et al., 2005*). We thus sought to explore whether EE could improve hippocampus-dependent learning in *Fmr1* KO mice. *Fmr1* KO mice and their WT littermates were subjected to a short-term EE paradigm as previously established (*Figure 3A*; *Hsu et al., 2019*). Briefly, animals were placed in the EE for 4–6 hr daily for a consecutive ten-day period. Fear conditioning training was conducted 1 day after the last EE exposure. Fear memory was examined using fear conditioning test and CA1 engram reactivation was measured by FOS staining (*Figure 3A*). Similar to home cage-raised mice, *Fmr1* KO mice showed normal freezing during learning when trained with the intensive fear conditioning paradigm (*Figure 3B*, *Figure 3—figure supplement 1A*). Moreover, EE experience fully reversed fear memory deficit in *Fmr1* KO mice as they exhibited normal freezing behavior during contextual memory recall 3 days after the training (*Figure 3C*). The rescue effect by EE in *Fmr1* KO mice learning was also apparent in the passive avoidance test done in a separate cohort of mice (*Figure 3D and E*). Nociception, working memory, locomotion, object recognition memory, and non-social anxiety levels were not affected by EE (*Figure 3—figure supplement 1B–F*). We next examined memory engram reactivation in *Fmr1* KO mice after EE experience. Corroborating the behavioral results, engram reactivation efficacy in *Fmr1* KO CA1 was also improved by EE to wild-type levels without significantly changing the overall CA1 activation levels during learning or memory recall (*Figure 3F–J*). Taken together, these results indicate that EE experience rescues fear memory deficits in the *Fmr1* KO mouse by improving the engram reactivation efficacy during memory recall.

## Neurons activated during enriched environment experience are more likely to become engram cells in subsequent learning

Enriched environment has long been proposed as a treatment strategy for improving cognitive ability in neurodevelopmental disorders. In the case of FXS, EE promoted mature spines growth in the visual cortex, rescued hyperactivity, and improved cognitive function in *Fmr1* KO mice (*Oddi et al., 2015*). In addition, EE combined with targeted treatment has been reported to promote cognitive improvements in FXS patients (*Winarni et al., 2012*), although the mechanism is not clear. We decided to explore this question by investigating how activating neurons during EE influences their behavior in subsequent learning. Our EE protocol consists a daily 4–6 hr exposure of EE for a consecutive 10 days (*Hsu et al., 2019*). To label EE-activated neurons, we injected 4-OHT right before EE experience on day 1 to capture all EE-activated neurons during the first day EE experience when we expected maximum activation due to novelty (*Figure 4A*). To maximize capturing, we performed two more follow-up injections on days 2 and 3 of EE, each right before the EE experience. EE extensively activated neurons in CA1, as roughly 25% neurons were labeled by tdTomato in both WT and KO groups (*Figure 4B and C*). Immunolabeling with FOS showed comparable degrees of neuronal activation in CA1 by 1 hr and 10 day EE experience in WT and *Fmr1* KO mice (*Figure 4—figure supplement 1A–C*), indicating that absence of FMRP does not affect neuron engagement during EE. After 10 days of EE, we trained the mice with conventional fear conditioning paradigm (to allow more precise FOS labeling of learning-activated neurons as the intensive training paradigm spans more than six hours) and examined CA1 engagement during learning with FOS staining. Overall CA1 engagement during fear learning was similar (~20%) between both WT and KO mice (*Figure 4B*

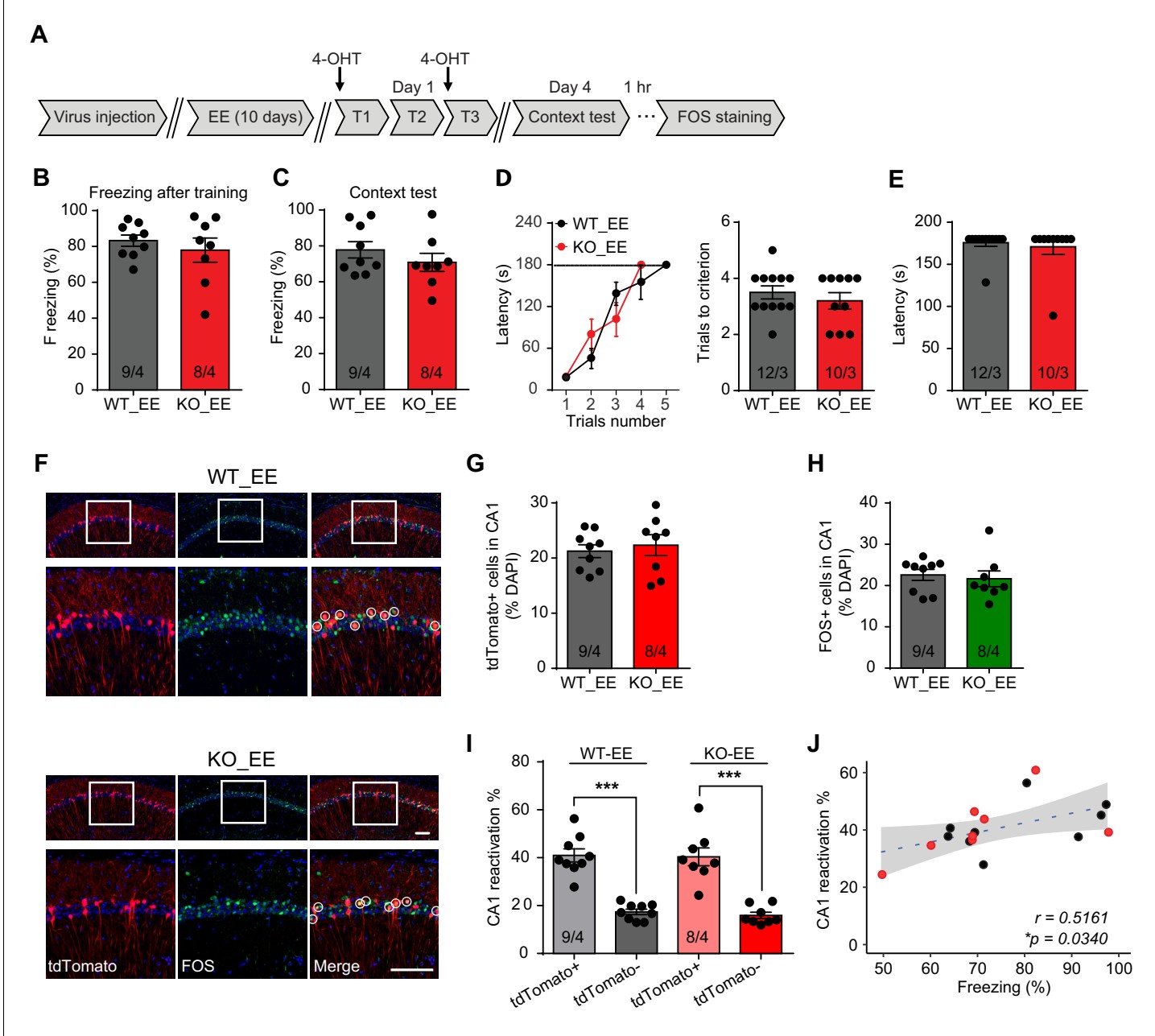

**Figure 3.** Enriched environment experience rescues fear memory deficits in *Fmr1* KO mice by improving engram reactivation efficacy during memory recall. (A) Experimental protocol for EE, memory-encoding neural ensemble and reactivation labeling. (B and C) Contextual fear conditioning test. Freezing levels were measured immediately after fear conditioning training (B) and again in training context 3 days after (C). (D and E) Passive avoidance test. (D) Learning curves (left) and trials number to criterion (right) were measured during training, and contextual fear memory in passive avoidance was measured 1 day later (E). (F) Representative images showing memory-encoding neural ensembles (engram cells) labeled with tdTomato (red) and memory recall-activated neurons labeled with FOS immunostaining (green). The circles in zoomed in images highlight reactivated neurons (yellow). Scale bar: 100 µm. (G) Quantification of percentage of neurons activated during learning. (H) Quantification of percentage of neurons activated during memory recall. (I) Quantification of engram reactivation in CA1. Percent FOS-positive neurons in both tdTomato+ and tdTomato- populations were measured [one-way ANOVA with Tukey's multiple comparison test: $F_{(3, 30)}=32.51$, $p<0.0001$, ***$p<0.001$]. (J) Positive correlation between engram reactivation efficacy (percent FOS+ neurons in tdTomato+ population) and behavioral performance during contextual memory test (*$p<0.05$, Pearson correlation coefficient). n/N, number of mice/number of independent litters. All graphs represent mean ± SEM.

The online version of this article includes the following figure supplement(s) for figure 3:

**Figure supplement 1.** Behavioral characterization of constitutive *Fmr1* KO mice with enriched environment experience (EE).

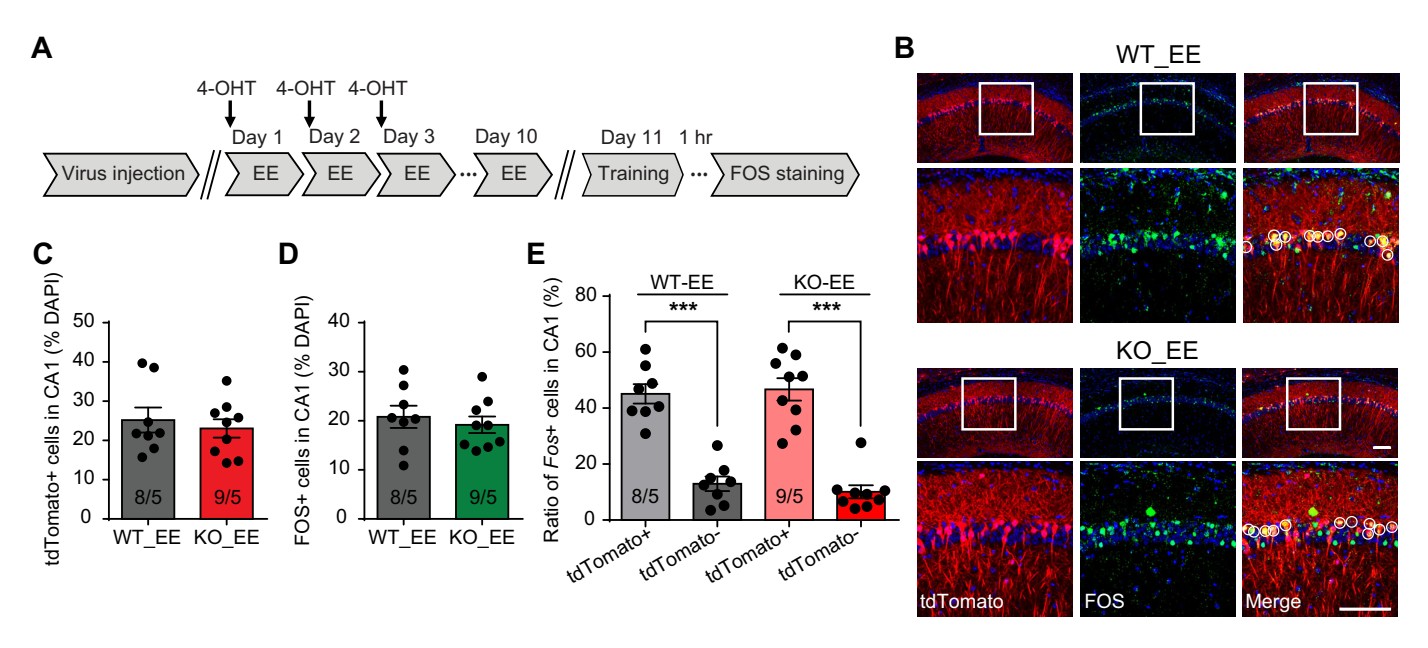

**Figure 4.** Neurons activated during enriched environment experience are more likely to become engram cells in subsequent learning. (**A**) Experimental protocol for enriched environment (EE) and neural ensemble labeling. (**B**) Representative images of EE-activated neural ensembles labeled with tdTomato (red) and fear conditioning training-activated neurons labeled with FOS immunostaining (green). The circles in zoomed in images highlight reactivated EE-engaged cells during fear conditioning learning (yellow). Scale bar: 100 μm. (**C**) Quantification of percentage of neurons activated during EE. (**D**) Quantification of percentage of neurons activated during fear conditioning learning. (**E**) Quantification of neuronal activation during subsequent learning (FOS+) in EE-engaged (tdTomato+) and non-EE engaged (tdTomato-) populations [Kruskal-Wallis test with Dunn's multiple comparison test: $H$ (3)=24.75, p<0.001, ***p<0.001]. n/N, number of mice/number of independent litters. All graphs represent mean ± SEM.

The online version of this article includes the following figure supplement(s) for figure 4:

**Figure supplement 1.** Histological characterization of constitutive *Fmr1* KO mice with enriched environment experience (EE).

---

*and D*). Interestingly, compared to tdTomato- neurons (not engaged in EE), which had very low FOS expression (~10%), tdTomato+ neurons (EE-activated) showed a four-fold increase in FOS expression, indicating that fear learning preferentially recruited EE-activated neurons (*Figure 4E*). This is the case for both WT and *Fmr1* KO CA1 (*Figure 4E*). Thus, a behaviorally relevant activity history (i.e. activation during EE) facilitates the likelihood of a neuron to participate in subsequent learning tasks.

## Inhibiting CA1 neurons activated by the EE experience prevents memory improvement by EE in *Fmr1* KO mice

The impact of EE on the brain is multi-faceted and involves multiple brain regions including cortex, hippocampus and cerebellum (*Leger et al., 2012*; *Nithianantharajah and Hannan, 2006*; *van Praag et al., 2000*). Based on our observation that CA1 EE-activated neurons are preferentially recruited to encode memory (*Figure 4E*), we further investigated whether EE's impact on CA1 network contribute critically to the rescue of *Fmr1* KO mice behavior. To answer this question, we applied a chemogenetic approach to specifically inhibit EE-activated CA1 neurons during fear conditioning learning. AAV-DJ-EF1a-DIO-hM4D(Gi)-mCherry was co-injected with AAV-*Fos*-TRAP to the CA1 region of Ai9 mice (WT or *Fmr1* KO), and 4-OHT was injected during EE to label and to express the inhibitory DREADD in EE-activated neurons (*Figure 5A*). CNO (5 mg/kg) was injected 1 hr prior to the fear conditioning training to specifically suppress activities of EE-activated neurons. Control mice received exact same treatment including CNO injections except no DREADD-expressing AAVs were injected. As expected, DREADD expression did not affect tdTomato labeling of EE-activated neurons in either WT or *Fmr1* KO mice (*Figure 5B and C*). Additionally, normal freezing behavior during training and similar levels of CA1 activation were observed in control and DREADD-expressing WT and *Fmr1* KO mice trained by conventional fear conditioning paradigm (*Figure 5B and D*, *Figure 5—*

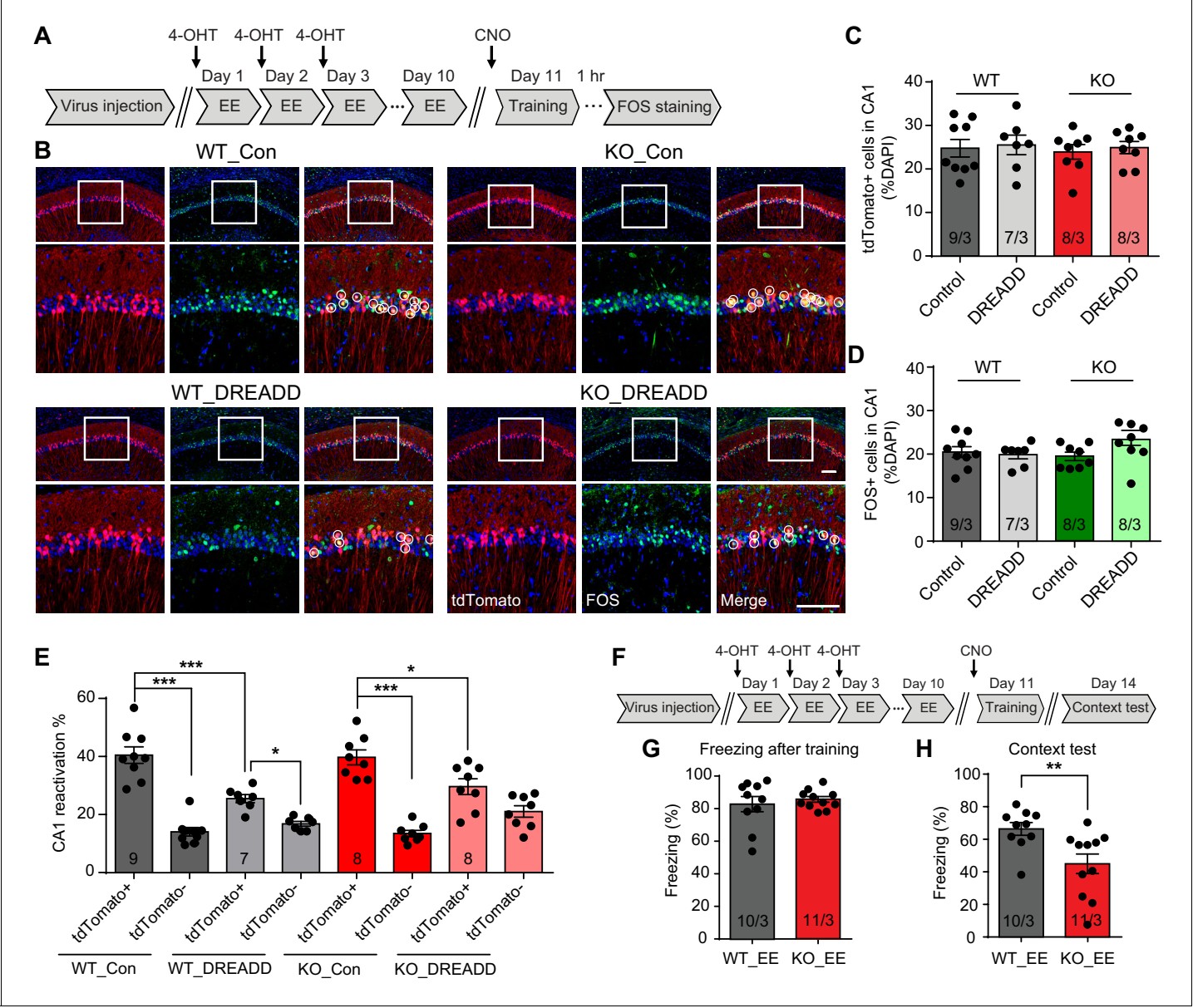

**Figure 5.** Inhibiting CA1 neurons activated by the enriched environment experience prevents memory improvement by EE in *Fmr1* KO mice. (A) Experimental protocol for chemogenetic inhibition of EE-activated neurons before fear conditioning training. (B) Representative images showing EE-activated neural ensembles labeled with tdTomato (red) and fear conditioning learning-activated neurons labeled with FOS immunostaining (green) in control and inhibitory DREADD-expressing mice CA1. The circles highlight reactivated EE-engaged cells during fear conditioning learning (yellow). Scale bar: 100 μm. (C) Quantification of neuronal activation by EE (Kruskal-Wallis test: *H* (3)=0.1518, p=0.9850). (D) Quantification of neuronal activation by fear conditioning learning (Kruskal-Wallis test: *H* (3)=5.669, p=0.1289). (E) Quantification of neuronal activation during subsequent learning (FOS+) in EE-engaged (tdTomato+) and non-EE engaged (tdTomato-) populations [WT group: two-way ANOVA: group factor, $F_{(1, 28)}$=9.608, **p<0.01; interaction, $F_{(1, 28)}$=20.22, ***p<0.001; Tukey post hoc test: ***p<0.001; *p<0.05. KO group: two-way ANOVA: group factor, $F_{(1, 28)}$=0.3268, p=0.5721; interaction, $F_{(1, 28)}$=16.29, ***p<0.001; Tukey post hoc test: ***p<0.001; *p<0.05]. (F) Experimental protocol for EE-activated neural ensemble inhibition and contextual fear memory test. (G and H) Contextual fear conditioning test. Freezing levels were measured immediately after fear conditioning training (G) and again in training context 3 days after (H) (**p<0.01, two-tailed Mann-Whitney *U* test). n/N, number of mice/number of independent litters. All graphs represent mean ± SEM.

The online version of this article includes the following figure supplement(s) for figure 5:

**Figure supplement 1.** Behavioral characterization of *Fmr1* KO mice after inhibition of EE-activated neurons in CA1.

*figure supplement 1A and B*). A closer look showed that DREADD expression in EE-activated neurons significantly dampened their participation in subsequent learning in both WT and *Fmr1* KO neurons, indicated by a significant shift of FOS+ neurons from the tdTomato+ population to the tdTomato- population (*Figure 5E*). As a result, the preferential recruitment of EE-activated neurons by learning is significantly reduced in both WT and *Fmr1* KO mice (the preference is not completely eliminated because inhibitory DREADD only induces modest hyperpolarization but does not completely shut down neuronal activity [*Roth, 2016*]). Taken together, the results so far suggest that when EE-activated neurons are inhibited by DREADD, new population of neurons that were not activated by EE (tdTomato-) could be engaged to support the demand of learning.

What happens to memory encoding in *Fmr1* KO mice if we prevent majority of EE-activated neurons from participating in learning? Using the same approach mentioned above, we tested memory recall 3 days after learning with EE-activated neurons being chemogenetically inhibited during learning (*Figure 5F*). DREADD-expressing WT and *Fmr1* KO mice showed normal freezing during and after intensive fear conditioning training (*Figure 5G*, *Figure 5—figure supplement 1C*). Interestingly, the context memory rescue observed in EE-experienced *Fmr1* KO mice was abolished by inhibiting EE engaged neurons, shown as significantly decreased freezing levels compared to wild-type mice (*Figure 5H*). CNO injection itself showed no effect to animal's locomotion (*Figure 5—figure supplement 1D*) or freezing behavior (*Figure 5—figure supplement 1E–G*). Inhibition of EE-activated neurons in wild-type mice have no significant effect on fear memory recall, indicated by comparable freezing levels during context memory test (p=0.1809 when comparing wild-type group freezing levels between *Figures 5H* and *3C*). Taken together, these results indicate that EE experience, through its direct impact on CA1 neurons, improves engram reactivation accuracy and rescues memory encoding in *Fmr1* KO mice.

## Discussion

Synaptic plasticity has long been thought as the cellular mechanism of learning and memory (*Hebb, 1949*). Neural ensemble theory proposed by Donald O. Hebb hypothesized that learned associations are stored in a population of neurons, which is achieved through synaptic plasticity development during and after learning. Reactivation of these neurons in the ensemble, often referred to as the engram cells, elicits memory recall (*Josselyn and Tonegawa, 2020*). Validation of the neural ensemble theory was made possible by the development of activity-dependent neuronal labeling technique (*Cowansage et al., 2014*; *Guenthner et al., 2013*; *Liu et al., 2012*; *Reijmers et al., 2007*). In these studies, optogenetic activation and inactivation of learning-activated neural ensembles leads to facilitation and impairment of memory recall, respectively (*Roy et al., 2016*; *Tanaka et al., 2014*), supporting the idea that memory traces are stored in engram cells and that their reactivation is critical for memory recall. In this study, we applied these approaches to investigate the mechanism underlying cognition deficits in the FXS mouse model. We show that successful reactivation of CA1 engram during memory recall test, which is essential for normal fear memory retrieval, is impaired in *Fmr1* KO mice, and that the recall efficacy is positively correlated with behavioral performances. Importantly, experiencing an enriched environment prior to learning improves engram reactivation efficacy and rescues memory recall in the *Fmr1* KO mice.

Both behavioral deficits and synaptic dysfunctions in the FXS animal models have been extensively investigated over the past decades (*Bostrom et al., 2016*; *Melancia and Trezza, 2018*; *Pfeiffer and Huber, 2009*; *Tian et al., 2017*). However, it remains unclear how altered synaptic plasticity, examined in ex-vivo systems (e.g. brain slices or cultured neurons), contributes to impaired memory encoding processes measured in behaving animals. In this study, we aim to further understand how network level changes, in particular, altered activation patterns of neuronal population during memory recall, contributes to memory deficits in the FXS mouse. Using well-established activity-dependent genetic labeling method, we showed that learning-activated neurons in hippocampal CA1 are indeed preferentially reactivated during memory recall, and that the efficacy of this engram reactivation is positively correlated with behavioral performances. In the *Fmr1* KO mice, reduced efficacy of neuronal ensemble reactivation correlates with reduced performance levels in fear memory recall test. What mechanism may contribute to the impaired neural ensemble reactivation in the *Fmr1* KO mice? Memory formation requires persistent changes at synapses that often involve new protein synthesis (*Ryan et al., 2015*). Given FMRP's function as an RNA-binding protein that

regulates the synthesis of many synaptic plasticity-related proteins (*Darnell and Klann, 2013*), and altered protein synthesis-dependent synaptic plasticity is a signature phenotype in the *Fmr1* KO mice (*Huber et al., 2002*), it is conceivable that the cognitive deficit observed in *Fmr1* KO mice is due to defective transformation of transient learning-induced neuronal activation into lasting memory traces. Indeed, our results show that despite the normal performance during learning, measured as freezing levels during and immediately after fear conditioning training and total number of neurons activated during learning and memory recall, *Fmr1* KO mice show impaired engram reactivation during memory recall. Thus, we concluded that the deficits in memory recall most likely occur during the period of memory consolidation after learning, when development of synaptic plasticity supports changes in network connectivity and memory trace formation. This is consistent with findings from a previous study showing that during spatial exploration, although hippocampal CA1 places cells in WT and *Fmr1* KO mice exhibit similar firing rates, those in FXS mice showed reduced specificity and impaired stability (*Arbab et al., 2018*).

How does EE experience improve fear memory formation in the *Fmr1* KO mice? Our results showed that not only EE rescued fear memory recall in *Fmr1* KO mice, but also restored engram reactivation accuracy in these mice, suggesting that the memory consolidation process was restored after EE. Multiple mechanisms have been proposed that may contribute to EE-mediated memory improvement, including enhanced neurogenesis (*Kempermann et al., 1997*), increased dendritic density and complexity (*Faherty et al., 2003*), changes in synaptic protein expression (*Hüttenrauch et al., 2016*; *Nithiananthrajah et al., 2004*) and neurotrophic factors (*Bekinschtein et al., 2011*). These changes may, directly or indirectly, lead to enhanced network function and synaptic plasticity. Additionally, enhanced sensory, motor and social experience during EE may support prolonged behaviorally meaningful patterns of neural activity, which may shift Hebbian plasticity rules. Deficits in Hebbian plasticity have been widely reported in FXS models (*Huber et al., 2002*; *Lauterborn et al., 2007*). mGluR-dependent LTD are enhanced, but not absent, in both hippocampus and cerebellum of *Fmr1* KO mice (*Hou et al., 2006*; *Huber et al., 2002*; *Koekkoek et al., 2005*). Studies investigating hippocampal and cortical LTP in *Fmr1* KO mice showed that LTP impairment depends on the age and strain of mice, and is induction protocol-dependent (*Lauterborn et al., 2007*; *Martin et al., 2016*; *Paradee et al., 1999*), largely suggesting that the LTP expression mechanism is intact in *Fmr1* KO mice, and that the observed deficits reflect an altered threshold for LTP induction (*Desai et al., 2006*; *Meredith et al., 2007*). The fact that EE can rescue fear conditioning performance in *Fmr1* KO mice supports the notion that basic signaling mechanisms underlying some forms of Hebbian plasticity (e.g. LTP) essential for fear memory encoding may be intact, but that the induction rules for these essential forms of plasticity have been changed in *Fmr1* KOs due to altered non-Hebbian plasticity (i.e. homeostatic synaptic plasticity) (*Sarti et al., 2013*; *Soden and Chen, 2010*; *Zhang et al., 2018*). Indeed, it has been shown that animals raised in an EE environment exhibit enhanced Hebbian plasticity (*Artola et al., 2006*; *Duffy et al., 2001*). The short EE protocol used in this study has also been shown to alter Hebbian plasticity threshold in the hippocampus (*Hsu et al., 2019*). Importantly, when the activity of EE-engaged neurons is dampened chemogenetically, the improvement in both engram activation accuracy and fear memory recall is abolished. Taken together, our results suggest that neuronal activation during EE experience is instrumental toward facilitating learning-dependent development of synaptic plasticity through modification of synaptic states and/or circuit properties.

One intriguing observation in this study is the preferential recruitment of EE-engaged neurons to fear memory-encoding ensembles, both in WT and *Fmr1* KO mice. Neuronal excitability has been shown as a key factor in memory allocation (*Yiu et al., 2014*; *Zhou et al., 2009*), EE-activated neurons may have higher excitability due to their engagement in EE (*Malik and Chattarji, 2012*; *Valero-Aracama et al., 2015*), and thus are more likely to participate in subsequent learning (*Lisman et al., 2018*; *Sekeres et al., 2010*). It is interesting that EE-induced bias in neuronal activation during learning is very similar between WT and *Fmr1* KO mice, suggesting that this process is independent of FMRP expression. How does this biased activation of neurons in learning improve memory encoding in the *Fmr1* KO mouse? We can envision two possibilities. First, through a cell-autonomous process, repeated and behaviorally relevant activation during EE induces changes in *Fmr1* KO neurons that facilitates development of synaptic plasticity during learning, allowing normal memory consolidation to occur. A second possibility is based on the observation that CA1 pyramidal neurons are heterogeneous in their development, gene expression patterns, anatomy, and function

(*Danielson et al., 2016*; *Mizuseki et al., 2011*; *Soltesz and Losonczy, 2018*). Thus, EE experience, due to its nature in promoting spatial navigation among other features, may preferentially activate a subset of CA1 neurons that are more effective in contextual memory encoding. Given our observation that EE-activated neurons are more likely to participate in subsequent learning, EE experience in *Fmr1* KO mice may result in a significant shift in the subtypes of CA1 neurons recruited during learning toward better contextual memory encoding, thus improving memory recall in the *Fmr1* KO mice. Further detailed analysis (i.e. transcriptome profiling) of EE-activated neurons are required to further distinguish these possibilities.

Memory engram allocation and reactivation are key areas under intense investigation in the field of learning and memory (*Josselyn and Tonegawa, 2020*; *Lisman et al., 2018*). It is generally agreed that memory recall requires accessing the neural ensembles that constitute the memory engrams (*Frankland et al., 2019*; *Tonegawa et al., 2015*). Agreeing with many studies conducted in several brain regions including hippocampus (*Denny et al., 2014*; *Kitamura et al., 2017*; *Nakazawa et al., 2016*; *Tanaka et al., 2014*; *Tayler et al., 2013*; *Zelikowsky et al., 2014*), our data support the notion that memory engram reactivation is a non-random process that occurs significantly above chance level of neural ensemble reactivation. However, these studies, including ours, also show that the overall reactivation percentage never approaches 100% (roughly 10–40%, depending on the region, labeling approaches and behavioral paradigms) (*Tanaka et al., 2014*; *Tayler et al., 2013*). Just as Hebb predicted (*Hebb, 1949*), reactivation of a fraction of memory ensemble is sufficient to produce full memory recall. Two possibilities may underly this seemingly 'imperfect' reactivation of memory engram. First, memory engram is flexible and dynamic. There is potential built-in redundancy in the mechanism to ensure full memory recall by a partial presentation of cues (or partial usage of available cues). Such redundancy ensures reliable behavioral responses in face of the less reliable sensory world (e.g. seasonal changes of scenery in the same location of the woods), thus increasing the chance of survival for the animals. Understanding how this redundancy is encoded without compromising memory specificity is one of the key questions in the field. A recent study examining the different aspects of hippocampal memory engram (*Sun et al., 2020*) provides an excellent example in which two different IEG-labeled ensembles act in a push-and-pull manner to ensure a balance between memory discrimination and generalization. A second possibility is that imperfect genetic labeling may contribute (at least partly) to the perceived imperfection of memory engram reactivation. So far, the immediate early gene promoters are the main approach to label engram cells. With a relatively broad labeling time window (ranging from a few hours to several days [*DeNardo and Luo, 2017*]) and varied IEGs induction profiles in different cell-types (*Cruz et al., 2013*; *Sun et al., 2020*), the current IEG promoter approach may 'over-tag' or 'under-tag' the engrams, contributing to imperfect alignment between neural ensemble activation during learning and its reactivation during memory recall. Nonetheless, these tools provide unprecedented opportunities for accessing memory engrams. Future studies with improved tools will bring us better understandings of the mechanisms underlying memory formation and recall.

## Materials and methods

**Key resources table**

| Reagent type (species) or resource | Designation | Source or reference | Identifiers | Additional information |
|---|---|---|---|---|
| Gene (*Mus musculus*) | Fmr1 | PubMed | 14265 | |
| Strain, strain background (*Mus musculus, male*) | *Fmr1* KO: FVB.129P2-Pde6b⁺ Tyr^{c-ch}Fmr1^{tm1Cgr}/J | The Jackson Laboratory | JAX: 004624 RRID:IMSR_JAX:004624 | |
| Strain, strain background (*Mus musculus, male*) | Ai9: B6.Cg-Gt(ROSA) 26Sor^{tm9(CAG-tdTomato)Hze}/J | The Jackson Laboratory | JAX: 007909 RRID:IMSR_JAX:007909 | |

*Continued on next page*

*Continued*

| Reagent type (species) or resource | Designation | Source or reference | Identifiers | Additional information |
|---|---|---|---|---|
| Strain, strain background (*Mus musculus*, male) | conditional *Fmr1* KO | *Zhong et al., 2018* | N/A | |
| Antibody | Mouse monoclonal anti-FMRP | DSHB | Cat#2F5-1, RRID:AB_10805421 | (1:2.5) |
| Antibody | Rabbit polyclonal anti-c-FOS | Millipore | Cat#ABE457, RRID:AB_2631318 | (1:500) |
| Antibody | Mouse monoclonal anti-CaMKII | Abcam | Cat# ab22609, RRID:AB_447192 | (1:500) |
| Antibody | Goat polyclonal anti-rabbit IgG secondary antibody | Thermo Fisher Scientific | Cat#A-11034, RRID:AB_2576217 | (1:200) |
| Antibody | Goat polyclonal anti-mouse secondary antibody (Cy3-conjugated) | Jackson ImmunoResearch Laboratories | Cat#115-165-146, RRID:AB_2338690 | (1:100) |
| Antibody | Goat polyclonal anti-mouse secondary antibody (Cy2-conjugated) | Jackson ImmunoResearch Laboratories | Cat#115-225-146, RRID:AB_2307343 | (1:200) |
| Other | AAV-DJ-Syn-mCre-GFP | *Hsu et al., 2019* | N/A | AAV virus expressing mCre-GFP |
| Other | AAV-DJ-Syn-Cre-GFP | *Hsu et al., 2019* | N/A | AAV virus expressing Cre-GFP |
| Other | AAV8-*Fos*-ER$^{T2}$-Cre-ER$^{T2}$-PEST | Gift from Karl Deisseroth lab *Ye et al., 2016* | N/A | AAV-*Fos*-TRAP |
| Other | AAV-DJ-EF1a-DIO-hM4D(Gi)-mCherry | Stanford Virus Core | Stock ID: AAV-129, Lot#5289 | AAV virus expressing Cre-dependent hM4D(Gi)-r |
| Other | AAV-DJ-EF1a-DIO-mCherry | Stanford Virus Core | Stock ID: AAV-14, Lot#5020 | AAV virus expressing Cre-dependent mCherry |
| Chemical compound, drug | 4-Hydroxytamoxifen (4-OHT) | Sigma-Aldrich | Cat#H6278 | 10 mg/ml |
| Chemical compound, drug | Clozapine N-oxide (CNO) | Tocris | Cat#4936 | 5 mg/kg |
| Chemical compound, drug | Corn oil | Sigma-Aldrich | Cat#C8267 | |
| Software, algorithm | Freezeview | Coulbourn Instruments | https://www.coulbourn.com/ | |
| Software, algorithm | FreezeFrame 4 | Coulbourn Instruments | RRID:SCR_014429 https://www.coulbourn.com/product_p/act-100a.htm | |
| Software, algorithm | Viewer III tracking system | Biobserve | RRID:SCR_014337 http://www.biobserve.com/behavioralresearch/products/viewer/ | |
| Software, algorithm | MED-PC IV | Med Associates | RRID:SCR_014296 https://www.med-associates.com/product/deluxe-shuttle-box-avoidance-chamber-package-for-mouse/ | |
| Software, algorithm | Nikon Elements | Nikon | RRID:SCR_014329 https://www.microscope.healthcare.nikon.com/products/software/nis-elements/nis-elements-advanced-research | |
| Software, algorithm | Prism 6 | GraphPad | RRID:SCR_002798 https://www.graphpad.com/scientific-software/prism/ | |

## Animals

Animal experiments were conducted following protocol approved by the APLAC at Stanford University. The *Fmr1* KO mice (stock number 004624) in the FVB background, *Ai9* reporter mice (stock number 007909) in the C57BL/6 background were obtained from The Jackson Laboratory (The Jackson Laboratory, Bar Harbor, ME). The *Fmr1* KO mice were back crossed to the C57BL/6 background for more than 10 generations before use. To obtain the wild-type or *Fmr1* KO mice for neural ensemble labeling, heterozygous females were crossed to *Ai9* reporter males. Since *Fmr1* is an X chromosome linked gene, the male littermates with *Fmr1* KO allele will be used as FXS animal, and littermates with WT allele will be used as wild-type control. Both FXS mice and wild type control have *Ai9* reporter allele. The conditional *Fmr1* KO mice in C57BL/6 background were generated as previously described (*Zhong et al., 2018*). Mice were group housed with littermates and maintained under a 12 hr/12 hr daylight cycle. Only male mice at age of 7–9 weeks old were used for all the experiments for the reasons listed below.

FMR1 is X-linked in both mice and humans. In both species, hemizygous *FMR1* loss-of-function leads to the severe symptoms in males due to a complete lack of FMRP expression, whereas females (heterozygous for *FMR1* mutation) exhibit a spectrum of abnormalities due to a mosaic expression of FMRP as a result of stochastic X-inactivation of one of the two alleles. Since the purpose of this study is to establish the pathophysiology caused by the *FMR1* loss-of-function, we decided to use only male mice to enable an unequivocal interpretation of phenotypes in hemizygous littermate WT and mutant male mice. To control for genetic background, all analyses were performed in littermates.

## AAV preparation and stereotaxic injection

AAVs were prepared as previously described (*Zolotukhin et al., 1999*). For *Fmr1* regionally deletion, the concentration of AAV-DJ-Syn-Cre-GFP and AAV-DJ-Syn-mCre-GFP were adjusted to $1.0 \times 10^9$ infectious units per ml. For neural ensemble labeling, AAV8 containing *Fos*-ERT2-Cre-ERT2-PEST was adjusted to $3.0 \times 10^{12}$ genomic copies (GC) per ml. For EE-neural ensemble inhibition, AAV-DJ-EF1a-DIO-hM4D(Gi)-mCherry and AAV8-*Fos*-ERT2-Cre-ERT2-PEST were adjusted to $3.0 \times 10^{12}$ genomic copies (GC) per ml and co-injected. AAV-DJ-EF1a-DIO-mCherry was adjusted to the same titer and used as control. All surgeries were performed under stereotaxic guidance. Mice were anesthetized using a mixture of ketamine and xylazine (80 mg/kg and 10 mg/kg, respectively). The virus was injected using a glass micropipette attached to a 10 μl Hamilton syringe. The pipette tips were beveled to be sharp and smooth. 0.60 μl of AAV were injected at a flow rate of 0.15 μl/min bilaterally. The coordinates used for CA1 were: 1.95 mm posterior to the bregma; 1.25 mm from the midline; 1.30 and 1.15 mm below the dura. We waited for 4 min both before and after each injection. AAVs were allowed to express for 2–3 weeks before experiment.

## Drug preparation

4-Hydroxytamoxifen (4-OHT, Cat#H6278; Sigma-Aldrich, St. Louis, MO) was dissolved at 20 mg/ml in ethanol by shaking at room temperature and was then aliquoted and stored at −20℃ for up to several weeks. The final 4-OHT solutions were always freshly prepared on the day of use. Before use, 4-OHT was re-dissolved in ethanol by shaking at room temperature for 30 min, corn oil (Cat#C8267, Sigma-Aldrich, St. Louis, MO) was added to give a final concentration of 10 mg/ml 4-OHT, and the ethanol was evaporated by vacuum under centrifugation. Final does of 4-OHT is 50 mg/kg and it was administrated right before fear conditioning training or enriched environment stimulation. Clozapine *N*-oxide (CNO, Cat# 4936; Tocris, Bristol, UK) was dissolved at 5 mg/ml in DMSO and the final does 5 mg/kg was administrated at 1 hr prior to the fear training. All injections were delivered intraperitoneally (i.p.).

## Enriched environment

Enriched environment (EE) was performed as described in our previous study (*Hsu et al., 2019*). Briefly, mice were placed in a Habitrail Ovo hamster cage (Hagen Inc, Mansfield, MA) filled with toys that vary in texture, size, shape, material, and color for 4–6 hr daily for a consecutive 10 days.

## Behavioral experiments

Mice were handled daily for 3 days before each behavioral test. Each mouse was subjected to either fear conditioning test or passive avoidance test, but never both. For all the other tests, each mouse was tested in the order of Y maze, open field, novel object recognition, elevated plus maze, and hot plate test with minimum 3 days interval between tests.

### Fear conditioning

Intensive fear conditioning training paradigm: for the intensive fear conditioning training protocol were applied in all neural ensemble labeling experiments to facilitate the efficacy of activity-dependent labeling. The protocol was slightly modified from a previously described protocol (*Liu et al., 2012*). The mice were singly housed at 3 days before fear training. On the training day, the mouse was placed in the fear conditioning chamber (Coulbourn Instruments, Allentown, PA) located in the center of a sound attenuating cubicle. The conditioning chamber was cleaned with 10% ethanol to provide a background odor. A ventilation fan provided a background noise at ~55 dB. The mouse received three training trials separated by 2 hr in their home cage. For each training trial, the mouse was kept in the conditioning chamber for 500 s. A tone (20 s, 85 dB, 2 kHz) was turned on at 180 s, 260 s, 340 s, and 420 s, each of which was co-terminated with a foot shock (2 s, 0.75 mA). After the third training trial, the mouse was returned to its home cage. Three days after training, the mouse was placed back to the original conditioning chamber for 5 min. The behavior of the mice was recorded with the FreezeFrame software and analyzed with the FreezeView software (Coulbourn Instruments, Allentown, PA). Motionless bouts lasting more than 1 s were considered as freeze. On the training day, the freezing percentages across the training period and after the third foot shock were summarized as an indication of fear memory acquisition.

Conventional fear conditioning training paradigm: on the training day, mice were placed in the fear conditioning chamber and allowed to explore it for 2 min. After exploration period, a tone (30 s, 85 dB, 2 kHz) was turned on at 120 s, 210 s, and 300 s, each of which was co-terminated with a foot shock (2 s, 0,75 mA). The mouse remained in the training chamber for another 30 s before being returned to the home cages. The next day after fear training, mice were placed back into the original conditioning chamber for 5 min. Mouse behavior recording and data analysis was the same as described above.

### Open field test

Mice were individually placed into the center of a 40 cm (L) x 40 cm (W) x 40 cm (H) open field chamber. Locomotor activity was recorded for 30 min using an overhead digital camera and tracked using Viewer III tracking system (Biobserve, Bonn, Germany). The center area was defined as the 20 by 20 cm central section of the chambers and the ratio of track length in the center area was used to estimate the anxiety level in the open field.

### Spontaneous alternation Y-maze

A light gray plastic Y-maze (Stoelting Co., Wood Dale, IL) was used to evaluate spatial working memory. Individual mice were placed in the center of the Y-maze and allowed to freely explore for 5 min. The sequences and total numbers of arm entries were recorded and analyzed with the Viewer III tracking system. Visiting all three different arms consecutively was termed a 'correct' trial and visiting one arm twice or more than three consecutive entries was termed a 'wrong' trial. Spontaneous alternation was calculated as the percentage of the 'correct' trial to the total trials.

### Elevated plus maze

The gray-painted maze had four $30 \times 8$ cm$^2$ arms. Two of them were open arms without walls and the other two were enclosed by 10 cm high walls. The maze was elevated 40 cm over the floor. At the beginning of the tests, mice were individually placed at the junction of an open and a closed arm, facing the closed arm. Then, the mice were allowed to freely move in the whole maze for 5 min. The time mice spent in the open versus the closed arms was recorded and analyzed with the Viewer III tracking system.

## Novel object recognition

The novel object recognition test was performed as described previously (*Leger et al., 2013*). Briefly, mice were given 10 min of habituation time individually in an empty box on day 1. On day 2, mice were individually placed in the box for 10 min with two identical objects, they are either T75 cell culture flask filled with bedding material or blocks of LEGO. On day 3, one object was replaced with a novel object. Object location in the chamber was randomized. Exploration behavior was recorded and analyzed with the Viewer III tracking system. The recognition index was defined as the time spent on the novel object divided by the time spent on both objects.

## Passive avoidance test

Passive avoidance test was performed using MED-PC IV software in a shuttle box (Med Associates, Fairfax, VT). The shuttle box was equipped with a metal grid floor and a door separating the box into equal halves. One half of the box was kept in the dark and the other half was exposed to a bright light. The mice were placed in the bright chamber and allowed to explore for 30 s, then the door was raised to allow the mice to enter the dark chamber. Once the mouse entered the dark chamber, the door was closed to prevent the mouse from escaping. After 3 s delay, a mild foot shock (0.3 mA) with 4 s duration was administered. Then the door was open, and the second training trial begins. The sequence was repeated until the mouse remained in the bright chamber for 180 s. The mice were given a single recall trial 24 hr after training. In the recall trials, the mice were placed in the lighted chamber for 180 s and the latency to enter the dark chamber was recorded.

## Hot plate test

Hot plate test was performed using a hot plate with custom floorless Plexiglas chamber (TAP Plastic, Mountain View, CA). The mice were placed on the hot plate maintained at 55°C, the latency to elicit a nocifensive behavior (e.g. hind paw withdrawal or licking) was recorded and the animal was promptly removed from the container. The maximum latency was set to 30 s to prevent tissue injury.

## Histology

Mice were deep anesthetized and perfused with 15 ml PBS followed by 15 ml ice cold fixative (4% paraformaldehyde diluted in PBS). The brains were postfixed in 4°C overnight and then immersed in 30% sucrose solution for 48 hr before being sectioned at 30 μm thicknesses on a cryostat (Leica Bio-systems, Wetzlar, Germany). For immunocytochemistry, free-floating mouse brain sections were per-meabilized with 0.3% Triton X-100 for 1 hr, then blocked for 1 hr in blocking buffer (5% FBS, 0.3% Triton X-100). Slices were subsequently incubated overnight with primary antibodies diluted in block-ing buffer, followed by 3 × 10 min washes in PBS and 2 hr incubation with corresponding fluores-cently labeled secondary antibodies in blocking buffer. The primary antibodies used included rabbit polyclonal anti-c-FOS (Millipore, Burlington, MA), mouse monoclonal anti-FMRP (Developmental Studies Hybridoma Bank, Iowa, IA), and mouse monoclonal anti-CaMKII (Abcam, Cambridge, UK). Secondary antibody included Goat polyclonal anti-rabbit IgG, Alexa Fluor 488 (Thermo Fisher Scien-tific, Waltham, MA), Cy2-conjugated goat polyclonal anti-mouse secondary antibody (Jackson Immu-noResearch Laboratories, West Grove, PA) and Cy3-conjugated goat polyclonal anti-mouse secondary antibody (Jackson ImmunoResearch Laboratories, West Grove, PA). Slices were then labeled with DAPI (Sigma-Aldrich, St. Louis, MO) diluted in PBS for 15 min, washed 3 × 10 min and mounted on glass slides with Fluoromount-G (Southern Biotech, Birmingham, AL).

## Imaging

All images were acquired using a Nikon A1 Eclipse Ti confocal microscope (Nikon, Tokyo, Japan) with a 20x objective, operated by NIS-Elements AR acquisition software. Laser intensities and acqui-sition settings were established for individual channels using optimal LUT settings and applied to entire experimental replicates. Images of three to five slices were taken per animal. Image analysis was conducted using Nikon Elements and all imaging and analyses were performed blind to the experimental conditions.

## Statistical analysis

All experiments were conducted with the experimenters being 'blind' to the genotypes and treatment parameters. Sample sizes (n number of independent biological replicates) were first determined based on similar experiments performed and published by our lab and others in the same field (*Hsu et al., 2019*; *Tanaka et al., 2014*; *Titley et al., 2017*). After experiments were completed, post hoc power analysis were performed to ensure minimal power of 0.8 is achieved at α value of 0.05. Additionally, to ensure reproducibility, each experiment was performed in at least three independent litters (N number), with multiple animals per litter. For non-behavioral experiments (e.g. histology), a minimum of three technical duplicates were analyzed and results averaged to generate a single data point for that particular animal. All samples in each group were analyzed without exclusions with one exception: if the injection site was found inaccurate during end-point histological verification, all data from that particular animal were excluded. The n/N (number of mice/number of independent litters) for each experiment are indicated in figure legends.

All results are presented as mean ± SEM, and statistical analysis were performed using GraphPad Prism six software (GraphPad Software, San Diego, CA). The distribution of data in each set of experiments was tested for normality using D'Agostino and Pearson omnibus normality test. Two-tailed unpaired *t* test (for parametric test) or Mann-Whitney *U* tests (for nonparametric test) was applied for two groups comparisons. For multiple groups comparisons, one-way ANOVA (for parametric test) or Kruskal-Wallis test (for nonparametric test) were applied, followed by appropriate post hoc tests (as indicated in figure legends). Animals in the same litter were randomly assigned to different treatment groups and blinded to experimenters. Injection sites and viral expression were confirmed for all animals, mice with incorrect injections sites were excluded from the data analysis.

## Acknowledgements

We thank Drs. Mu Zhou and Omid Miry for helpful discussions throughout the study. The work was supported by NIH grants MH086403 (LC), NS11566001 (LC), HD084215 (LC), a postdoctoral fellowship from the Stanford Maternal and Child Health Research Institute (JL), and Stanford Undergraduate Advising and Research Grants (RYJ).

## Additional information

### Competing interests

Lu Chen: Reviewing editor, *eLife*. The other authors declare that no competing interests exist.

### Funding

| Funder | Grant reference number | Author |
| --- | --- | --- |
| National Institute of Mental Health | MH086403 | Lu Chen |
| Eunice Kennedy Shriver National Institute of Child Health and Human Development | HD084215 | Lu Chen |
| National Institute of Neurological Disorders and Stroke | NS11566001 | Lu Chen |
| Eunice Kennedy Shriver National Institute of Child Health and Human Development | 1P50HD104458 | Lu Chen |
| Stanford Maternal and Child Health Research Institute | | Jie Li |

The funders had no role in study design, data collection and interpretation, or the decision to submit the work for publication.

## Author contributions
Jie Li, Conceptualization, Data curation, Formal analysis, Funding acquisition, Validation, Investigation, Writing - original draft, Writing - review and editing; Rena Y Jiang, Data curation, Formal analysis, Funding acquisition, Validation; Kristin L Arendt, Yu-Tien Hsu, Sophia R Zhai, Data curation, Formal analysis; Lu Chen, Conceptualization, Resources, Supervision, Funding acquisition, Investigation, Writing - original draft, Project administration, Writing - review and editing

## Author ORCIDs
Jie Li [ID] https://orcid.org/0000-0001-7009-5680
Lu Chen [ID] https://orcid.org/0000-0002-8097-2699

## Ethics
Animal experimentation: The study was performed in strict accordance with the recommendations in the Guide for the Care and Use of Laboratory Animals of the National Institutes of Health. All of the animals were handled according to approved institutional animal care and use committee (IACUC) protocols (#28021) of Stanford University. All surgery was performed under Ketamine hydrochloride and Xylazine anesthesia, and every effort was made to minimize suffering.

## Decision letter and Author response
Decision letter https://doi.org/10.7554/eLife.61882.sa1
Author response https://doi.org/10.7554/eLife.61882.sa2

# Additional files
## Supplementary files
• Transparent reporting form

## Data availability
All data generated and analyzed in this study are included in the manuscript.

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
