## [Decision Letter]

**Acceptance summary:**

This study examines the hippocampal "memory engram" of contextual fear conditioning in *Fmr1* knockout mice. The authors find memory consolidation is impaired with the loss of *Fmr1* and this impairment is associated with a reduction in the re-activation of the hippocampal engram. This finding provides a circuit-level link for previous research on the synaptic impacts and behavioral deficits causes by loss of *Fmr1*. The authors further demonstrate that environmental enrichment is an effective way to restore the cognitive function of the mutant mice, and this restoration is accompanied by the restoration of the reactivation of the hippocampal engram.

**Decision letter after peer review:**

Thank you for submitting your article "Defective memory engram reactivation underlies impaired fear memory recall in Fragile X syndrome" for consideration by *eLife*. Your article has been reviewed by three peer reviewers, and the evaluation has been overseen by a Reviewing Editor and Laura Colgin as the Senior Editor. The following individuals involved in review of your submission have agreed to reveal their identity: Jeansok Kim (Reviewer #2); Anton Maximov (Reviewer #3).

The reviewers have discussed the reviews with one another and the Reviewing Editor has drafted this decision to help you prepare a revised submission.

There was strong enthusiasm for the study reporting the reactivation of hippocampal CA1 engram cells essential for normal fear memory retrieval is impaired in the *Fmr1* KO mouse model. The reviewers request further explanation on the data as to freezing to tone and the interpretation of behavioral results. The reviewers also commented on further mechanistic insights into the overlap between EE-activated neurons and memory engrams (e.g. add electrophysiology experiments), however this is not required and a further elaboration of potential mechanisms would also be acceptable. We ask that you address the following comments listed below.

Essential revisions:

1) Both intensive fear conditioning and conventional fear conditioning training procedures comprised of 12 tone-shock pairings and 3 tone-shock pairings, respectively. However, only contextual freezing data are presented. Some explanation is needed as to why tone freezing data are not included. Also, the authors may want to consider the possibility that the reduced contextual freezing behavior observed in *Fmr1* KO mice may not necessarily reflect impaired contextual fear memory but rather stronger discriminative fear conditioning to the tone CS, which is explicitly paired with the shock US. In other words, if the tone CS accrued more associative strength, the context fear will become weaker.

2) In the Discussion, the authors offer potential explanations for an intriguing recruitment of EE-activated cells for subsequent memory coding but, at the end, the underlying mechanism remains mysterious. While extensive molecular and structural characterization of these cells goes beyond the scope of the present study, the hypothesis that EE-activated cells are more excitable could be easily tested via current-clamp recordings in acute brain slices.

3) In Figure 2B, considerable numbers of cFos-positive neurons appear to be GABAergic interneurons as their somas reside outside of CA1sp. This seems inconsistent with quantifications of the overlap between cFos- and CamkII-immunoreactive cells, as assessed with a different reporter.

4) The authors should improve the presentation and labeling of data shown in Figure 2 and Figure 2—figure supplement 1. The term HC is confusing (home cage?), and there seems to be a discrepancy between Figure 2C and Figure 2—figure supplement 1D in terms of percentages of recombined neurons.

5) The rationale for multiple injections of 4-OHT is unclear.

6) The study clearly shows that only ~40% of neurons recombined during learning become cFos-positive after memory recall, suggesting that engram ensembles are flexible. The authors should further emphasize this point in the Discussion. Also, they should discuss recent studies demonstrating that cFos and cFos-based TRAP systems may not label the full spectrum of neurons essential for memory storage (see Sun et al., 2020).

7) It is important to explain why only male mice were used in the main text since many readers may not be familiar with sex-related differences in the manifestation of FXS.

8) The manuscript will benefit from additional proof reading. For example:

– Introduction: "involved in regulation of"

– Figure 1 legend: Panel A is mistakenly referred to as panel B

– Figure 2 legend: "I" in "Intensive" should be capitalized

– Rephrase "Each mouse was subjected to only one behavioral test for fear conditioning test and passive avoidance test" to "Each mouse was subjected to either fear conditioning test or passive avoidance test."

– Insert spaces "The protocol was slightly modified from apreviously(Liu et al., 2012).…"

---

## [Author Response]

Essential revisions:1) Both intensive fear conditioning and conventional fear conditioning training procedures comprised of 12 tone-shock pairings and 3 tone-shock pairings, respectively. However, only contextual freezing data are presented. Some explanation is needed as to why tone freezing data are not included. Also, the authors may want to consider the possibility that the reduced contextual freezing behavior observed in Fmr1 KO mice may not necessarily reflect impaired contextual fear memory but rather stronger discriminative fear conditioning to the tone CS, which is explicitly paired with the shock US. In other words, if the tone CS accrued more associative strength, the context fear will become weaker.

We thank the reviewer for pointing this out. We adopted the conventional delayed fear conditioning paradigm in our study for the purpose of relating our study to the vast literature, which includes both contextual and cued fear conditioning components. We did not test the tone-freezing in our original experiments because we needed the FOS labeling to reflect neuronal activation during contextual fear memory recall. However, we want to mention that in our original experiments using CA1-specific *Fmr1* KO mice, we did also measure the cued fear memory in response to tone and found that it was also impaired. This data is now shown Author response image 1. To further specifically address this concern in the constitutive *Fmr1* KO mouse, we have performed another set of experiments in three independent cohorts where cued fear memories were measured. Similar to our previous experiments, both the WT and *Fmr1* KO mice exhibited normal learning curve during training (Author response image 1). By contrast, the cued memory to tone examined three days after learning was significantly impaired in *Fmr1* KO mice (Author response image 1). Taken together, these results indicate that cued fear memory is also impaired in *Fmr1* KO mice.

The equally impaired tone freezing response partially addressed the second point raised by this reviewer in that the weaker context fear is not the result of discriminative fear conditioning because the tone fear memory is not stronger or normal. To further address this point with a more direct approach, we trained another three independent cohorts of mice with a modified intensive training protocol by leaving out the tone CS. Again, both WT and *Fmr1* KO mice learned normally during training (Author response image 1 and H). However, contextual fear memory is still impaired in the *Fmr1* KO group (Author response image 1), indicating that independent of the presence of tone CS, contextual fear memory is indeed impaired in FXS mouse model. This result is not unexpected as *Fmr1* KO mice also exhibited impairment in passive avoidance test, which is similar to contextual fear memory (please refer to manuscript Figures 1C and D).

**Author response image 1. sa2fig1:** Cued fear memory and contextual fear memory are both impaired in CA1-*Fmr1* KO mice and constitutive *Fmr1* KO mice. (A and B) Altered context test and cued fear memory test in CA1-*Fmr1* KO mice. Freezing levels in an altered context (A) and altered context with tone CS (B) were measured three days after training (**, *p* < 0.01, two-tailed Mann-Whitney *U* test). (C-F) Cued fear memory in constitutive *Fmr1* KO mice. (C) Learning curves of constitutive *Fmr1* KO mice in cued fear conditioning. The intensive fear conditioning training paradigm has three sessions, each consists of four pairs of co-terminating tone (20 s, gray bar) and foot shock (2 s, red bar). Freezing levels before, during and after each tone-shock pair were quantified (Two-way repeated measure ANOVA: training 1: group factor, *F*(1, 11) = 3.497, *p* = 0.0883; interaction, *F*(24, 264) = 1.776, *, *p* < 0.05; training 2: group factor, *F*(1, 11) = 14.90, **, *p* < 0.01; interaction, *F*(24, 264) = 1.150, *p* = 0.2899; training 3: group factor, *F*(1, 11) = 2.745, *p* = 0.1258; interaction, *F*(24, 264) = 1.632, *, *p* < 0.05). (D) Freezing levels measured immediately after fear conditioning training. (E) Freezing levels in an altered context and (F) altered context with tone CS were measured three days after training (**, *p* < 0.01, two-tailed Mann-Whitney *U* test). (G-I) Contextual fear memory in constitutive *Fmr1* KO mice. (G) Learning curves of constitutive *Fmr1* KO mice in contextual fear conditioning. The intensive fear conditioning training paradigm has three sessions, each consists of four foot shock (2 s, red bar) delivered at 198 s, 278 s 358 s and 438 s. Freezing levels before, during and after each shock were quantified (Two-way repeated measure ANOVA: training 1: group factor, *F*(1, 12) = 1.497, *p* = 0.2446; interaction, *F*(24, 288) = 0.8571, *p* = 0.6612; training 2: group factor, *F*(1, 12) = 0.4586, *p* = 0.5111; interaction, *F*(24, 288) = 0.7814, *p* = 0.7597; training 3: group factor, *F*(1, 12) = 0.1740, *p* = 0.6839; interaction, *F*(24, 288) = 0.8450, *p* = 0.6775). Freezing levels were measured immediately after fear conditioning training (H) and again in training context three days after (I) (**, *p* < 0.01, two-tailed Mann-Whitney *U* test). n/N, number of mice/number of independent litters. All graphs represent mean ± SEM.

2) In the Discussion, the authors offer potential explanations for an intriguing recruitment of EE-activated cells for subsequent memory coding but, at the end, the underlying mechanism remains mysterious. While extensive molecular and structural characterization of these cells goes beyond the scope of the present study, the hypothesis that EE-activated cells are more excitable could be easily tested via current-clamp recordings in acute brain slices.

We thank the reviewer for the suggestions and agree it would be interesting to further investigate the underlying mechanism that drives recruitment of EE-activated cells. Excitability changes are indeed one of the potential mechanisms that biases the recruitment toward EE-activated cells. In agreement with the hypothesis this reviewer brought up, previous studies performed in acute brain slices have already shown that EE experience increases intrinsic excitability of hippocampal CA1 cells (Malik and Chattarji, 2012; Valero-Aracama, Sauvage, and Yoshida, 2015), although it was not examined specifically in EE engaged neurons. Considering the phenomenon that neurons with relatively higher intrinsic excitability win the allocation competition to become engram cells (Lisman et al., 2018; Sekeres et al., 2010), we believe neuronal excitability increases resulting from EE experience could largely contribute to the subsequent memory engram allocation. To investigate this mechanism further with comprehensive electrophysiology analysis in acute brain slices is indeed our goal for the next step. In addition to excitability changes, we would also examine synaptic strength and synaptic plasticity changes induced by EE experience.

3) In Figure 2B, considerable numbers of cFos-positive neurons appear to be GABAergic interneurons as their somas reside outside of CA1sp. This seems inconsistent with quantifications of the overlap between cFos- and CamkII-immunoreactive cells, as assessed with a different reporter.

We thank the reviewer for his/her astute observations and apologize for accidently choosing a non-representative image in Figure 2B. Our quantification indeed shows that more than 94% of FOS-positive cells (shown as tdTomato reporter in the new experiment mentioned below) are CaMKII-positive non-GABAergic cells. We have replaced the images with another example that better represents this quantification.

To address the reviewer’s concern of characterizing engram cell types using different reporters, we re-performed this experiment using Ai9 reporter mouse line so that the engram cells were labelled with the same reporter (tdTomato). The results of CaMKII immunostaining in these hippocampal slices are now shown in updated Figure 2—figure supplement 1E and F. The percentages of CaMKII-positive neurons in tdTomato-labeled engram cells are 94.8% and 96.2% in WT and *Fmr1* KO groups, respectively, which are consistent to our previous results using EYFP as the reporter, and the other report using a similar labeling approach (Guenthner et al., 2013).

4) The authors should improve the presentation and labeling of data shown in Figure 2 and Figure 2—figure supplement 1. The term HC is confusing (home cage?), and there seems to be a discrepancy between Figure 2C and Figure 2—figure supplement 1D in terms of percentages of recombined neurons.

We apologize for the unintended confusion. We have replaced the ‘HC’ labeling in Figure 2—figure supplement 1 with ‘basal activity’.

5) The rationale for multiple injections of 4-OHT is unclear.

We have added the following explanation to the main text to further elaborate on the rationale behind our multiple 4-OHT injection for both learning activated engram labeling and EE engaged neurons labeling.

“Learning-activated neurons were labeled with 4-OHT injection. Since the effective time window of 4-OHT is around 6 hours after injection (Guenthner et al., 2013), and the intensive training protocol spans more than six hours, we performed two 4-OHT injections at the beginning of the first and third training sessions to ensure sufficient labeling (Figure 2A).”

“To label EE-activated neurons, we injected 4-OHT right before EE experience on day 1 to capture all EE-activated neurons during the first day EE experience when we expected maximum activation due to novelty (Figure 4A). To maximize capturing, we performed two more follow-up injections on day 2 and 3 of EE, each right before the EE experience”.

6) The study clearly shows that only ~40% of neurons recombined during learning become cFos-positive after memory recall, suggesting that engram ensembles are flexible. The authors should further emphasize this point in the Discussion. Also, they should discuss recent studies demonstrating that cFos and cFos-based TRAP systems may not label the full spectrum of neurons essential for memory storage (see Sun et al., 2020).

This reviewer brought up a very interesting point, which we too have been pondering about. Agreeing with many studies conducted in several brain regions including hippocampus (Denny et al., 2014; Kitamura et al., 2017; Nakazawa et al., 2016; Tanaka et al., 2014; Tayler et al., 2013; Zelikowsky et al., 2014), our data support the notion that memory engram reactivation is a non-random process that occurs significantly above chance level of neural ensemble activation. However, these studies, including ours, also show that the overall reactivation percentage never approaches 100% (roughly 10-40%, depending on the region, labeling approaches and behavioral paradigms) (Tanaka et al., 2014; Tayler et al., 2013). Just as Hebb predicted (Hebb, 1949), reactivation of a fraction of memory ensemble is sufficient to produce full memory recall.

What may underlie this seemingly ‘imperfect’ reactivation of memory engram? We consider two possibilities. First, imperfect genetic labeling may contribute to the perceived imperfection of memory engram reactivation. So far, the immediate early gene promoters are the main approach to label engram cells. With a relatively broad labeling time window (ranging from a few hours to several days (DeNardo and Luo, 2017)) and varied IEGs induction profiles in different cell-types (Cruz et al., 2013; Sun et al., 2020), the current IEG promoter approach may ‘over-tagging’ or ‘under-tagging’ the engrams during training or testing. Additionally, IEG promoter-driven Cre expression may not lead to expression of reporters in all cells where this IEG is activated, contributing to imperfect alignment between genetic labelling and direct IEG staining. Second, as the reviewer alluded to, memory engram is flexible and dynamic. There is potential built-in redundancy in the mechanism to ensure full memory recall by a partial presentation of cues (or partial usage of available cues). Such redundancy ensures reliable behavioral responses in face of the less reliable sensory world (e.g., seasonal changes of scenery in the same part of the woods), thus protecting the survival of the animals. Understanding how this redundancy is encoded without compromising memory specificity is one of the key questions in the field. The study by Sun et al., 2020, is an excellent example in which two different IEG-labeled ensembles act in a push-and-pull manner to ensure a balance between memory discrimination and generalization.

We have added the above-mentioned points in the Discussion.

7) It is important to explain why only male mice were used in the main text since many readers may not be familiar with sex-related differences in the manifestation of FXS.

We thank the reviewer for this suggestion. The following statement has been added to the Materials and methods section under “Animals”.

“FMR1 is X-linked in both mice and humans. In both species, hemizygous FMR1 loss-of-function leads to the severe symptoms in males due to a complete lack of FMRP expression, while females (heterozygous for FMR1 mutation) exhibit a spectrum of abnormalities due to a mosaic expression of FMRP as a result of stochastic X-inactivation of one of the two alleles. […] To control for genetic background, all analyses were performed in littermates. “

8) The manuscript will benefit from additional proof reading. For example:– Introduction: "involved in regulation of"– Figure 1 legend: Panel A is mistakenly referred to as panel B– Figure 2 legend: "I" in "Intensive" should be capitalized– Rephrase "Each mouse was subjected to only one behavioral test for fear conditioning test and passive avoidance test" to "Each mouse was subjected to either fear conditioning test or passive avoidance test."– Insert spaces "The protocol was slightly modified from apreviously(Liu et al., 2012).…"

We thank the reviewer’s careful reading and corrections. All changes have been made accordingly.